# A New Method to Map Spring Irrigated Areas Using MODIS LST Products and Ancillary Data in an Agricultural District of Northwest China

Yizhu Lu [1,2], Wenlong Song [1,2,*], Linjing Tian [3], Xiuhua Chen [1,2], Rongjie Gui [3] and Long Chen [1,2]

1   Sate Key Laboratory of Simulation and Regulation of Water Cycle in River Basin, China Institute of Water Resources and Hydropower Research, Beijing 100038, China
2   Research Center on Flood & Drought Disaster Reduction of the Ministry of Water Resources, Beijing 100038, China
3   College of Resource Environment and Tourism, Capital Normal University, Beijing 100048, China
*   Correspondence: songwl@iwhr.com; Tel.: +86-010-6878-5451

**Abstract:** Irrigation alleviates drought in croplands and maintains or increases crop yields. The accurate monitoring of irrigated areas is important to regional water resource management, food security, climate change, drought monitoring, and emergency disaster relief. Based on field experiments that demonstrate the feasibility of irrigated area mapping using land-surface temperature, we propose a method to map spring irrigation areas using historical meteorological data, main crop phenological characteristics, irrigation regimes, and MODIS land-surface temperature (LST) products. The distribution of irrigation intensity, spring irrigated areas (SIA, considering the irrigation intensity), and total area of spring irrigation (STIA, regardless of irrigation intensity) were monitored by the proposed method for the Donglei Irrigated District (Phase II) in northwestern China from 2011 to 2018. The spring irrigation of the study area was divided into three periods (16 January–23 February, 24 February–24 March, and 25 March–31 May). Then, the temperature threshold of the irrigated area in each period was determined by the diurnal temperature range (DTR) of the rain-fed plots and precipitation data; for the three periods, this was 12 °C, 15 °C, and 11 °C, respectively. The results showed that most of the croplands in the study area were irrigated once or twice. The SIA in most years varied between 55,900 and 73,100 ha, but in 2016, the irrigation area reached 100,200 ha. The STIA accounted for 60–70% of the irrigated area except 2016. The average accuracy of SIA monitoring was satisfactory and above 94% for years when sufficient and reliable data was available.

**Keywords:** irrigated area; spring irrigation; remote sensing; MODIS LST; arid regions; semi-arid regions





## 1. Introduction

Irrigation is estimated to use nearly 75% of all diverted freshwater resources to water approximately 17% of all croplands worldwide and to produce around 40% of all global food products [1]. Moreover, irrigation water use is expected to increase due to population growth and the need to increase agricultural production [2–4]. Irrigation practices can play an important role in maintaining or increasing crop yields under drought conditions [5]. Accurate information regarding the irrigated area and its spatial extent and variation are needed in order to manage the distribution of water resources and crop structure adjustments in irrigated districts, and for drought monitoring, and emergency mitigation [6–8]. Remote sensing monitoring of a wide range of irrigated areas enables us to monitor irrigation status and drought conditions in irrigated districts and provides data that can support decision making regarding use of water resources.

The first digital global irrigated area map was generated in 1999 by combining information from large-scale irrigated areas per country for the year 1995 and national data on total irrigated area per country, drainage basin, or federal state [9]. More recently, satellite remote

sensing has been effectively utilized for the large-scale monitoring of irrigated areas [10]. Scholars from all over the world have applied different methods to identify irrigated areas: agricultural census data [9], supervised and unsupervised classification [11,12], vegetation indices [13–16], statistical analysis [15,16], spectral matching techniques [17–21], machine learning algorithms [22–26], and other methods [6,7,27–29]. The International Water Management Institute (IWMI) published a 10-km spatial resolution global irrigated area map using a spectral matching technique based on satellite images of the Normalized Difference Vegetation Index (NDVI) [19]. Subsequent studies have used higher spatial resolution satellite imagery to generate more detailed irrigated area maps for Asia and Africa [16]. Recently, Xu et al. [26] derived a high spatial resolution (30-m) map of the irrigated area for southwestern Michigan (a humid to sub-humid region) by integrating a wide range of remote sensing images and hydroclimate information using a machine-learning algorithm. However, most of these techniques are based on vegetation indices to produce irrigated area maps. In fact, changes in the vegetation index are related not only to irrigation, but also to soil productivity, fertilizer application, crop type, and sowing time [29]. In addition, there is a lag between irrigation and vegetation growth, which can lead to uncertainty regarding the use of vegetation indices to determine the irrigated area. Furthermore, these approaches require data from a growing season or from a longer time series to establish an irrigated area map and can only generate the annual irrigated area or the total area available for irrigation but cannot monitor the irrigated area in near-real-time.

In order to overcome these limitations, we propose a more direct approach with a shorter cycle time to produce irrigated area maps during spring at a 1-km spatial resolution with a combination of data from land-surface temperature (LST) standard products retrieved from Moderate Resolution Imaging Spectroradiometer (MODIS), and ancillary data for the historical meteorological data and main crop phenological characteristics. We obtained four products: (1) Spring Irrigated Areas (SIA); (2) Irrigation intensity; (3) Total Irrigated Area in Spring (STIA); and (4) near-real-time Irrigation Area (NIA). We demonstrated this approach for the Donglei Irrigated District (Phase II). In addition, we also evaluated the accuracy of irrigation mapping using census-based statistics.

## 2. Materials

### 2.1. Study Area

The Donglei Irrigation District (Phase II) extends from 109°10′ E to 110°10′ E and from 34°41′ N to 35°00′ N. The district is located in the east of the Guanzhong region in Shaanxi Province, China (Figure 1). The Donglei Irrigation District (Phase II) is a large irrigation district and an important grain-producing district in the northwest of China, which has played an important role in supporting and ensuring the agricultural economy of Shaanxi province. The district has an average annual water consumption of 62.27 million m$^3$ [30]. The northwestern part of this region is higher than the southeastern part, with altitudes ranging from 385 m to 600 m. There are distinct dry and rainy seasons in the district; more than 50% of the rainfall is concentrated from July to September, while the annual precipitation ranges from 519 mm to 552 mm, and the annual evaporation ranges from 1700 mm to 2000 mm. These characteristics designate the region as semi-arid. Therefore, irrigation is the main source of water for crops in the district, especially in the spring. Irrigation in the study area is mainly through inundation via channels from the Yellow River. The designed irrigation area covers 84,333 ha and is divided into seven systems: Dali, Sunzhen, Pucheng, Xingzhen, Jingyao, Liuqu, and Liuji.

The main crops in the study area are winter wheat and summer corn, while other fruit and vegetable crops are also planted in a few areas. One or two crops may be planted in the study area in one year, and irrigation may occur in the winter, spring, or summer. Winter irrigation lasts from November to December, while spring irrigation begins from the middle of February and lasts to April's end. Summer irrigation takes place from early June to the end of August. The growth cycles of winter wheat are from October to June's middle of the subsequent year, and irrigation is usually performed once (in November or

March–April) or twice (in November and March–April). The growth cycles of summer corn are from late June to October, and irrigation is generally completed 2–3 times in June and July, with an interval of about one month between irrigation.

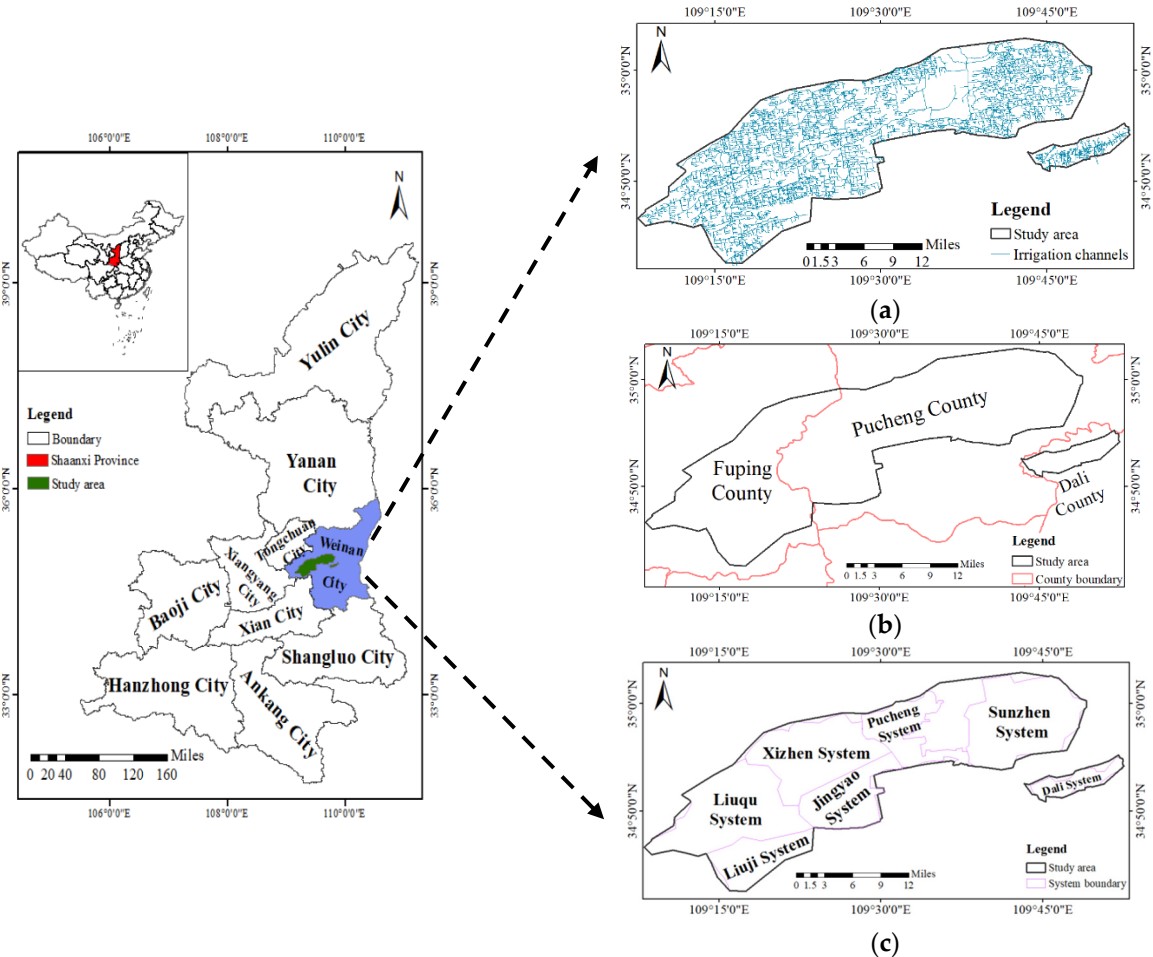

**Figure 1.** The location of the Donglei Irrigation District (Phase II). (**a**) canal distribution, (**b**) county distribution, (**c**) irrigation system distribution.

This section is divided by subheadings. It should provide a concise and precise description of the experimental results, their interpretation, as well as the experimental conclusions that can be drawn.

### *2.2. Data*

#### 2.2.1. MODIS LST Products

In this study, we used the daily temporal resolution of the MODIS LST product MOD11A1 to achieve near-real-time monitoring of the irrigated area [31]. MOD11A1 at a nominal pixel spatial resolution of 1 km can provide daytime and nighttime LST products and calculate the diurnal temperature range (DTR) [32]. The data were downloaded from the National Aeronautics and Space Administration Goddard Space Flight Center for the period from 16 January to 5 May, for the years 2011 to 2018. We used the quality control files that come with MOD11A1 products to filter out the data of good quality and eliminate the data with null values and poor quality. When the pixels with good quality were less than 30% of the study area, we defined that the image quality was low. We downloaded the MOD11A1 of the study area for the period from 16 January to 5 May, for the years 2011 to 2018. After removing the images with null values, we obtained images with available data, and performed quality control processing to classify images with

available data as good quality and low quality. Then, the annual quality of MOD11A1 in the study area was obtained (Figure 2).

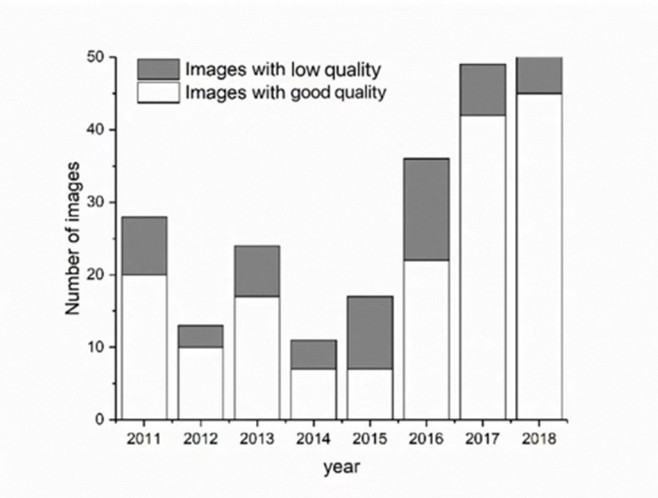

**Figure 2.** The diurnal temperature range (DTR) data (MOD11A1) during spring irrigation from 2011 to 2018 (16 January–5 May).

### 2.2.2. Meteorological Data

Most of the study area is located in Pucheng County and Fuping County, Weinan City, Shaanxi Province. Therefore, historical meteorological data, including the weather conditions, daily maximum temperature, daily minimum temperature, and DTR (https://m.tianqi.com/lishi/shanxi1, accessed on 15 March 2019) from 16 January to 5 May 2011–2018 in Pucheng County and Fuping County, were taken as the basis for the classification of spring irrigation periods and calibration thresholds.

### 2.2.3. Irrigated Area Statistics

The statistical data obtained from the irrigation district management department included SIA, start and end time of irrigation for each year from 2011 to 2018. These data were mainly used for the analysis, accuracy evaluation, and rationality analysis of the generated irrigated area maps.

### 2.3. LST Ground Monitoring Experiment

U.S. RAYTEK ST80+ infrared thermometer was used to monitor the daytime and nighttime LST of different underlying surfaces with different irrigation conditions for about 5 days to verify the rationality and feasibility of irrigated area mapping based on LST. The monitoring time was from 14 August to 19 August 2019. During the period, monitoring was carried out at 2 am and 2 pm every day to record the daily maximum and daily minimum surface temperature, air temperature, weather conditions, and irrigation conditions. The monitoring targets were water bodies, bare soil, corn, beans, and trees. On 15 August, parts of the bare soil, corn, and trees were irrigated to compare and analyze the impact of irrigation on LST.

### 3. Methods

Given the influence of solar radiation, ground feature compositions, physical states, thermal characteristics, geometric structures, ecological environments, soil physical parameters, and other factors, the LST and its DTR of various ground features are significantly different, such as sandy land, grassland, woodland, and lake [33]. Water bodies have the largest specific heat capacity (i.e., the temperature of water bodies increases/decreases less per unit of heat energy absorbed) among the common land cover features.

Therefore, water bodies heat up slowly during the day and cool slowly at night; the temperature change between day and night occurs more slowly and is smaller in bodies of water than the temperature change in surrounding land cover features. For areas with bare soil, the specific heat capacity of dry soil is far less than that of a water body; the dry bare soil heats up rapidly during the day and releases heat rapidly at night. Therefore, the temperature changes greatly from day to night, and the DTR is far greater than the DTR of a water body. Moreover, given that moisture can increase the specific heat capacity of bare soils, the DTR of moist bare soil is smaller than that of dry bare soil; essentially, the greater the moisture content in bare soil, the greater the specific heat capacity of the bare soil [34–36]. For areas with vegetation coverage, such as grasslands, woodlands, and croplands, the LST obtained by satellite remote sensing is influenced by vegetation and soil. Soil water content affects the evapotranspiration of vegetation, and evapotranspiration affects the surface temperature of vegetation [37]. When crops are short of water during the day, soil water cannot meet the necessary requirements for crop transpiration. Thus, transpiration and heat consumption are reduced, and the leaf temperatures rise, resulting in higher surface temperatures than those observed for crops without water shortages. Similarly, the surface temperature of water-deficient crops decreases faster at night than that of non-water-deficient crops due to the thermal insulation effect of water. Therefore, the DTR of water-deficient crops is greater than that of non-water-deficient crops [38,39]. The soil moisture content in croplands is obviously increased after irrigation, while the DTR is notably smaller than that of the non-irrigated cropland. Hence, it can be determined whether the cropland is irrigated or not by determining a DTR threshold. Irrigation can be considered to have occurred when the DTR is less than the threshold.

### 3.1. DTR Thresholding

Each pixel in the MOD11A1 images represents an approximate area of 1 km$^2$. However, irrigation is carried out by households in China, the area of cropland per household is small, and the irrigated areas are scattered. Consequently, it is difficult to find pixels that represent areas that have completely been irrigated in order to determine DTR thresholds. However, water from the Yellow River is used for irrigation through gravity flow channels, and a relatively large area exists that cannot be irrigated due to the high terrain (namely the rain-fed plots), which can be easily identified and used to determine the DTR thresholds. We conducted field investigations and collected longitude and latitude information for 24 rain-fed plots in June 2018. These plots were located in 7 pixels of the MODIS LST products (Figure 3). Areas that received precipitation were assumed to have similar image characteristics to irrigated areas, and areas that did not receive precipitation were assumed to have similar image characteristics to non-irrigated areas.

### 3.2. Decision Tree Irrigation Model

According to the determined thresholds, the distribution of the near-real-time irrigated area in the study area could be monitored after excluding the influence of precipitation by using the decision tree (Figure 4), namely, the spatial distribution maps of daily irrigated area during irrigation (Near-real-time Irrigation Area, NIA). As shown in Figure 4, historical meteorological data of Pucheng County and Fuping County were used to exclude the data affected by precipitation. The image data for days when there was precipitation and for the first day after precipitation were excluded. Image data for the second day to the fifth day after precipitation were judged in comparison to the image data on the first day after precipitation. If non-irrigation was present on the first day after precipitation in a certain region, the region showed irrigation on the next day or in the following few days after precipitation. This would indicate that the region on that day was not affected by precipitation, and so it did not need to be removed. Similarly, if irrigation occurred on the first day after precipitation in a certain region, the region showed irrigation and the DTR gradually increased within the second day to the fifth day after precipitation, indicating that the region was affected by precipitation and needed to be removed. After removing

the influence of precipitation, the distribution of NIA during the spring irrigation in the study area was monitored using the DTR threshold.

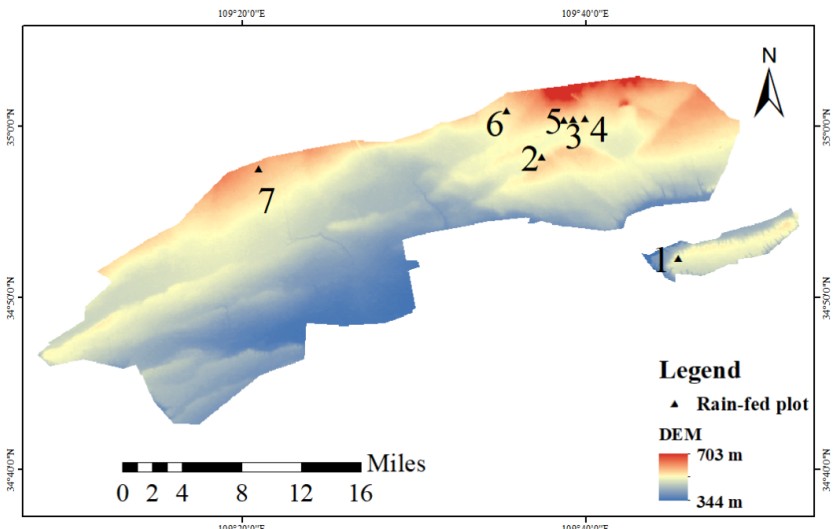

**Figure 3.** Digital Elevation Model (DEM) of the study region and the distribution of rain-fed plots in the study region. (The numbers in the figure represent the 7 pixels which 24 rain-fed plots located).

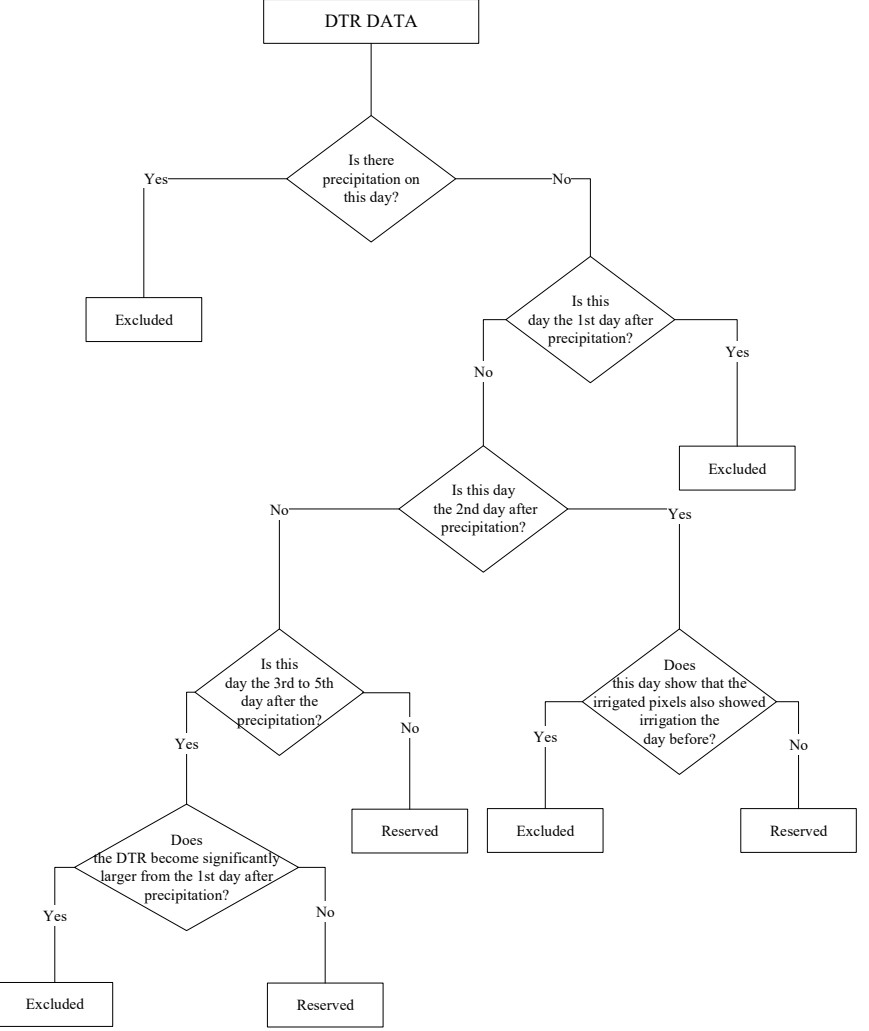

**Figure 4.** Decision tree classification excluding data affected by precipitation.

The interval between the two irrigation times in the study area was about 1 month, but a pixel cell of 1 km resolution is coarse, and the irrigation time in different areas within a pixel may be different, so the same pixel may appear many times in the monitoring results of one irrigation cycle. In addition, the cropland in the study area is inundated, and the soil water content of the cropland may remain high for several days after irrigation. Therefore, the decision tree was used in the calculation of SIA (considering the irrigation intensity, summing the area of each round of irrigation during the spring irrigation) and irrigation intensity (Figure 5). The irrigation cycle was 30 days (the first irrigation was used to indicate the start time of the cycle). If the same pixel appeared many times in a cycle, it was only considered one irrigation, and the SIA and irrigation intensity were calculated according to this principle. The total irrigated area in the spring (STIA), regardless of irrigation intensity, did not sum the areas that were irrigated multiple times, i.e., pixels where irrigation was detected more than once during the spring irrigation were. only counted once. This information was obtained through the NIA.

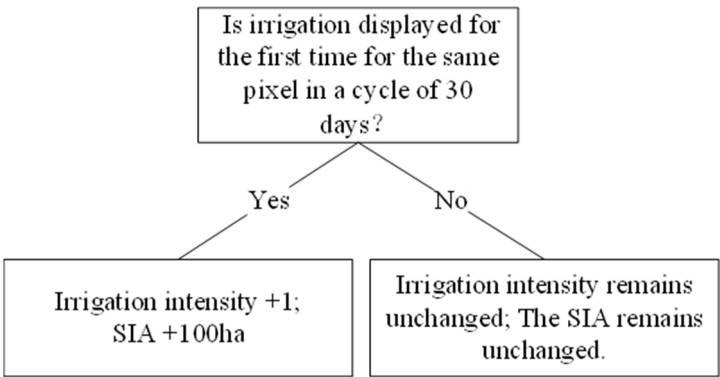

**Figure 5.** Decision tree for calculating the SIA and irrigation intensity.

## 4. Results

### 4.1. Rationality Analysis of Irrigated Area Mapping Based on LST

The daytime and nighttime LST monitoring results of different ground objects such as water bodies, corn, and beans are shown in Figure 6. The results show that the surface temperature of water bodies was the most stable and the temperature difference between day and night was the smallest. Vegetation coverage affected the surface temperature of ground objects. Under the same conditions, the higher the vegetation coverage, the smaller the daytime LST and DTR. After irrigation, the DTR of bare soil, corn, and walnut decreased, and the DTR of bare soil changed the most and fluctuated significantly, while corn and walnut with high vegetation coverage were less affected by irrigation. The experimental results are consistent with the theory that irrigation reduces the diurnal range of surface temperature. For features with lower vegetation coverage, this effect was more obvious. In the same environment, the vegetation coverage was high, the DTR was small, and the variation of surface temperature caused by irrigation was not as obvious as that with low vegetation coverage. Therefore, the irrigated area could be identified by the DTR, and this method was more suitable for targets with low vegetation coverage. The effect of irrigation measures on LST and its changes was analyzed through field experiments, and the feasibility of irrigated area mapping using LST was verified, which was used as a theoretical basis for studying the irrigated area mapping method based on LST.

### 4.2. DTR Threshold Determination

Surface temperature is affected by solar radiation and is positively correlated with air temperature changes, and this trend is particularly pronounced in the spring [40]. As a result, the temperature change during the spring irrigation can reflect a change in the LST trend to a certain extent. According to the analysis of the temperature and weather conditions in the spring of 2011–2018 (Figure 7), the daily maximum temperature and

the daily minimum temperature in the period from the beginning of January to the end of May showed a continuous warming trend. The variation trends of daily temperature ranges were positively correlated with the daily maximum temperature. After 23 February, the minimum temperature was generally above 0 °C, and after 25 March, the maximum temperature was generally above 15 °C.

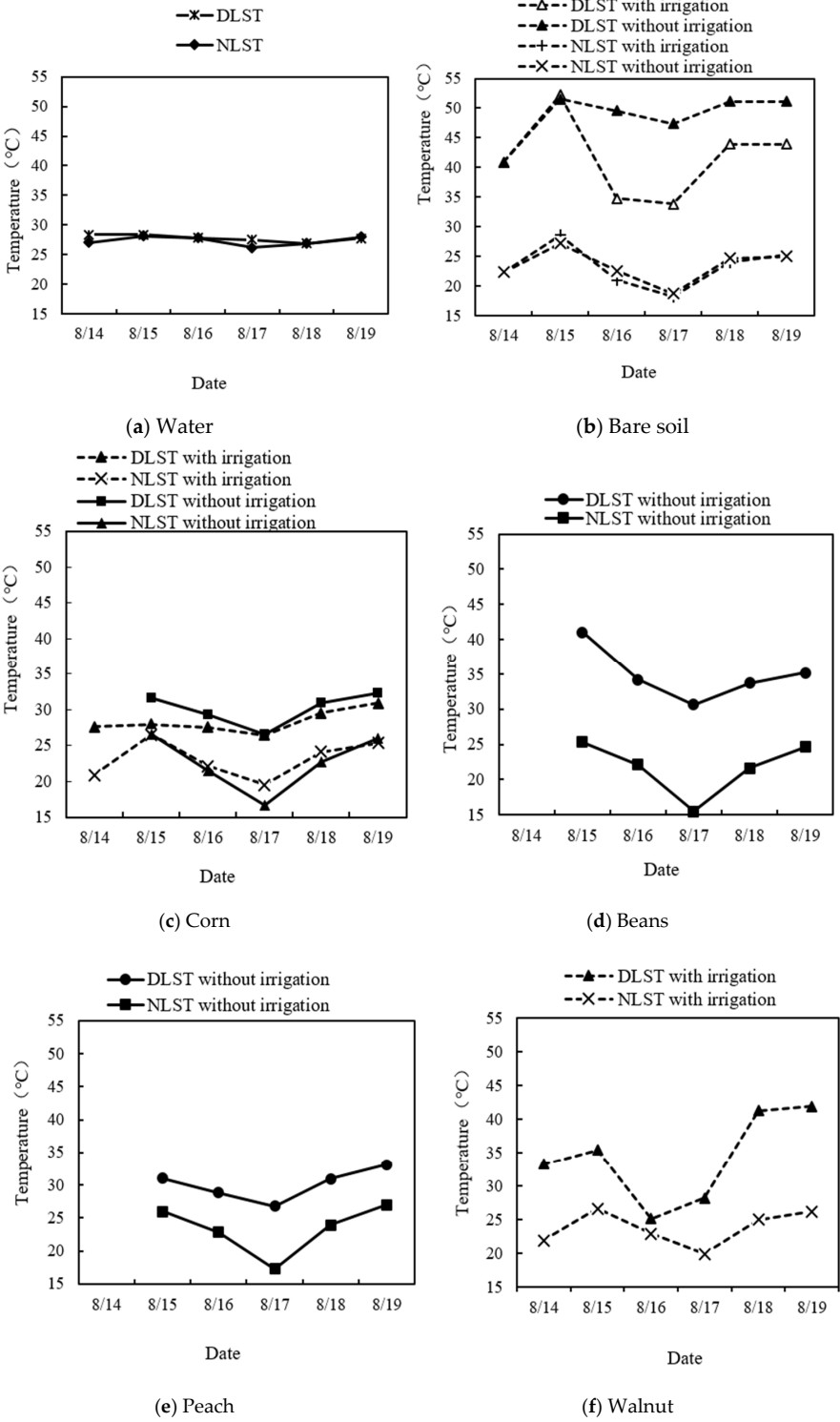

**Figure 6.** Daytime and nighttime LST monitoring results (DLST is short for daytime land surface temperature, NLST is short for nighttime land surface temperature). (**a**–**f**) are DLST and NLST monitoring results of water, bare soil, corn, beans, and peach respectively.

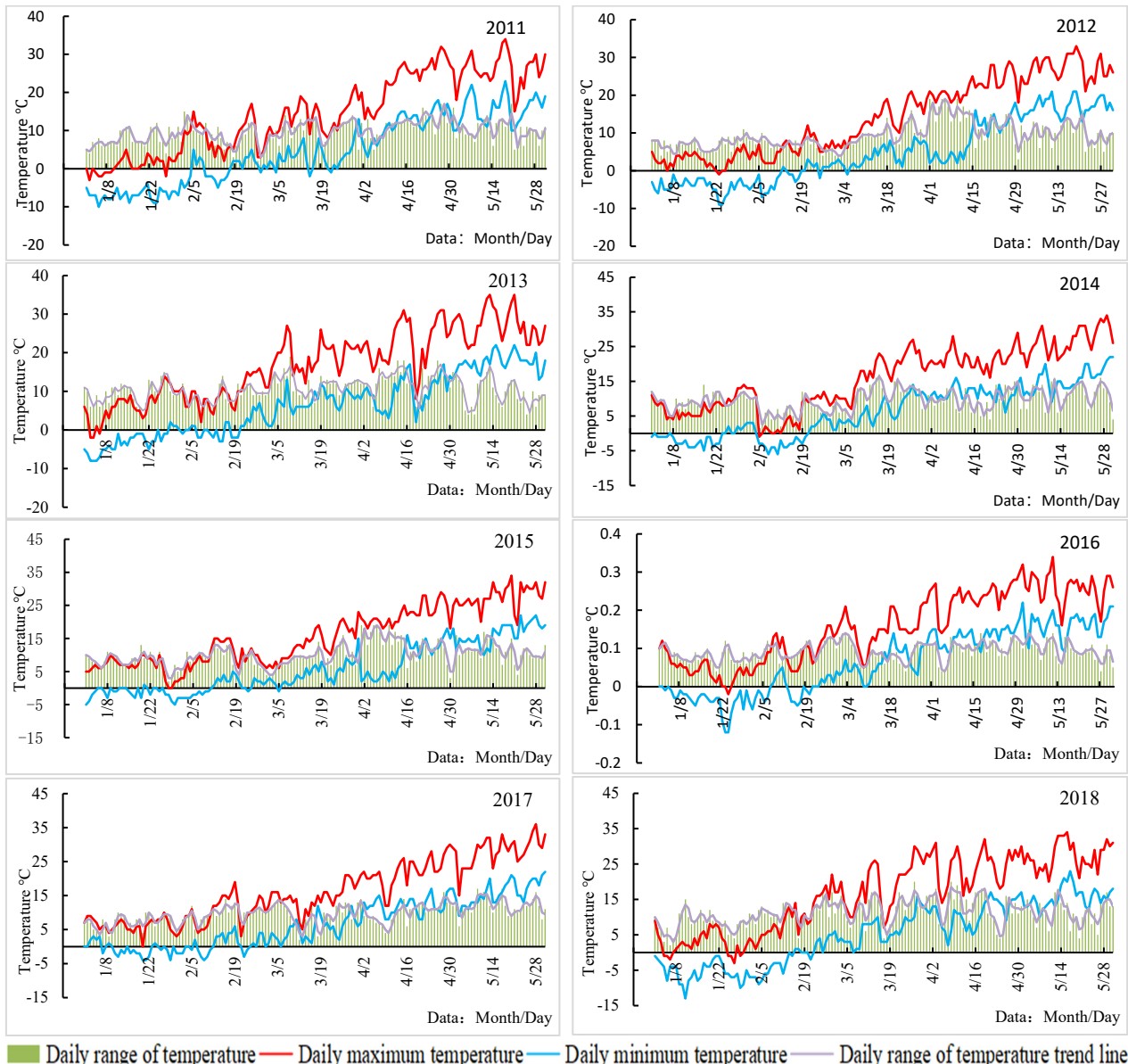

**Figure 7.** Daily maximum temperature, daily minimum temperature, and daily temperature ranges from January to May in 2011–2018.

Since the spring irrigation in the study area was mainly to irrigate the winter wheat crops, the winter wheat was from the revival stage to the filling stage during this period, which is the process of gradually increasing the vegetation coverage. As shown in Table 1, before the booting stage, the vegetation coverage rate was lower, and there was barer soil. At the booting stage and in later stages, the winter wheat plants grew rapidly, the vegetation coverage rate increased, and the amount of bare soil decreased [41].

According to the analysis results of historical meteorological data and the phenological characteristics of the main crops in the spring irrigation, the spring irrigation was divided into three periods: 16 January–23 February (the daily minimum temperature was generally below 0 °C, small plants, more bare soil), 24 February–24 March (the daily minimum temperature was generally above 0 °C and the daily maximum temperature was below 15 °C, small plants, more bare soil), and 25 March–31 May (the daily maximum temperature was generally above 15 °C, the plant growth was fast, vegetation coverage became greater, plants became larger, and the amount of bare soil decreased).

**Table 1.** Characteristics of winter wheat phenology in the study area.

| Growth Cycle | Emergence | Overwintering | Revival Stage | Jointing Stage | Booting Stage | Heading and Flowering Stage | Filling Stage | Maturity and Harvest Time |
|---|---|---|---|---|---|---|---|---|
| Data | October–November | December–January | early February–early March | middle to late March | late March–early April | middle to late April | early and middle May | late May–early and middle June |
| Characteristics | The first true leaf reveals the coleoptile | Stop growing | The wheat seedlings turn bright green with some new leaves emerging | The first internode of the main stem is 1.5–2 cm above the ground | This stage is mainly the growth of wheat branches and leaves, and the growth rate is fastest. | The top of the ear shows leaf sheath | The fastest period in which dry matter is stored in a grain | The water content of the plant decreases, the leaf color changes from green to yellow, and the grain turns yellow and hard |

Combined with the precipitation data and the DTR of the rain-fed plots, the threshold of the irrigated area in each period during the spring irrigation was determined. According to the statistical data of irrigation in the study area, the earliest starting time of spring irrigation was 20 January, and the latest ending time was 1 May. Therefore, the DTR from MODIS LST and the corresponding precipitation data-determined thresholds were used from 16 January to 5 May 2011–2018. The DTR data at rain-fed plots in the eight years were divided into three parts:

- The influence of precipitation data (i.e., data on the first day after precipitation). Evaporation was less on the first day after precipitation, which was more consistent with the cropland conditions after irrigation;
- Deleted data (i.e., data on the day of precipitation and from the second to the fifth day after precipitation). The temperature drop on the day of precipitation will affect the LST. From the second to the fifth day after precipitation, LST may be affected by precipitation, but the effects are incomplete, and the difference is different from the amount of irrigation water.
- Data that were not affected by precipitation.

The data affected by precipitation and not affected by precipitation in the three periods were sorted in ascending order. The 95% and 5% confidence levels were used to eliminate the MODIS data itself in order to identify outliers, and the threshold ranges were obtained. Combined with the historical meteorological data and according to the analysis, it was determined that the thresholds for identifying irrigation in the three periods of spring irrigation were 12 °C, 15 °C, and 11 °C, respectively.

In order to analyze the rationality of the threshold, we took the first period (16 January–23 February) of spring irrigation in 2018 as an example to compare the changes in the monitoring results of the irrigation area under different threshold conditions. As shown in Figure 8, STIA first increased with the increase in the threshold, then became relatively stable, and then increased suddenly. We used the abrupt point as the threshold where the curve stabilized and then increased. The truth of STIA during the first period of spring irrigation in 2018 was 33,500 ha, which was used for the accuracy calculation in Figure 8.

### 4.3. Distribution of the Spring Irrigated Area in 2011–2018

The threshold for period 1 (16 January–23 February) was 12 °C, the threshold for period 2 (24 February–24 March) was 15 °C, and the threshold for period 3 (25 March–5 May) was 11 °C. According to these thresholds, the distribution of irrigation districts in the spring irrigation of the study area in 2011–2018 was monitored in near-real-time after removing the data affected by precipitation. The SIA, STIA, and irrigation intensity of the study area were obtained (Figures 9 and 10, Table 2).

The irrigated area remote sensing monitoring method based on DTR could monitor the spatial distribution of the NIA in the study area, taking the period from 15 April 2018 to 18 April 2018 as an example (Figure 9). During this period, irrigation was mainly carried out in the southwestern portion of the study area, which clearly reflected the irrigation

process, the irrigation sequence in the different districts, and the spatial distribution of daily irrigation area.

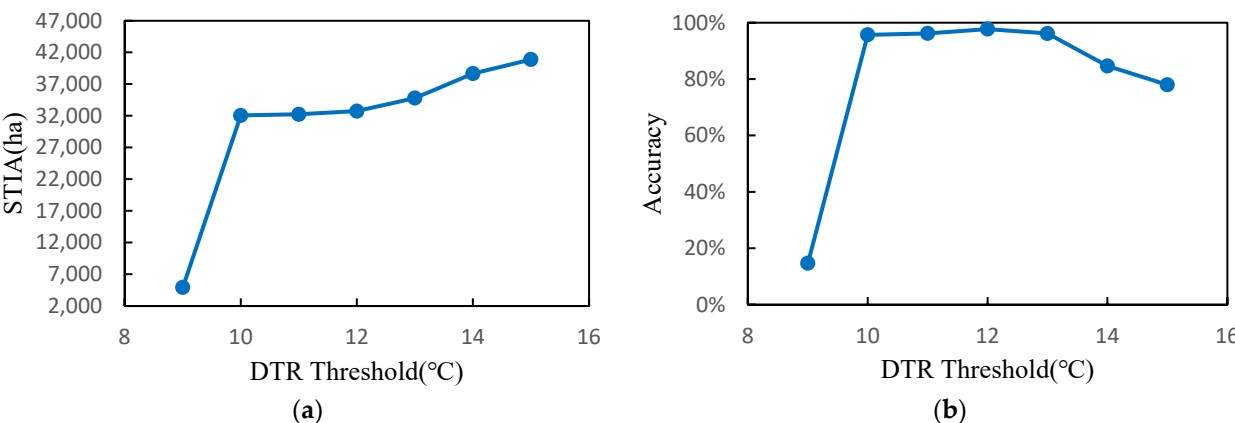

**Figure 8.** Sensitivity analysis of the DTR Threshold. (**a**) threshold sensitivity, (**b**) threshold accuracy.

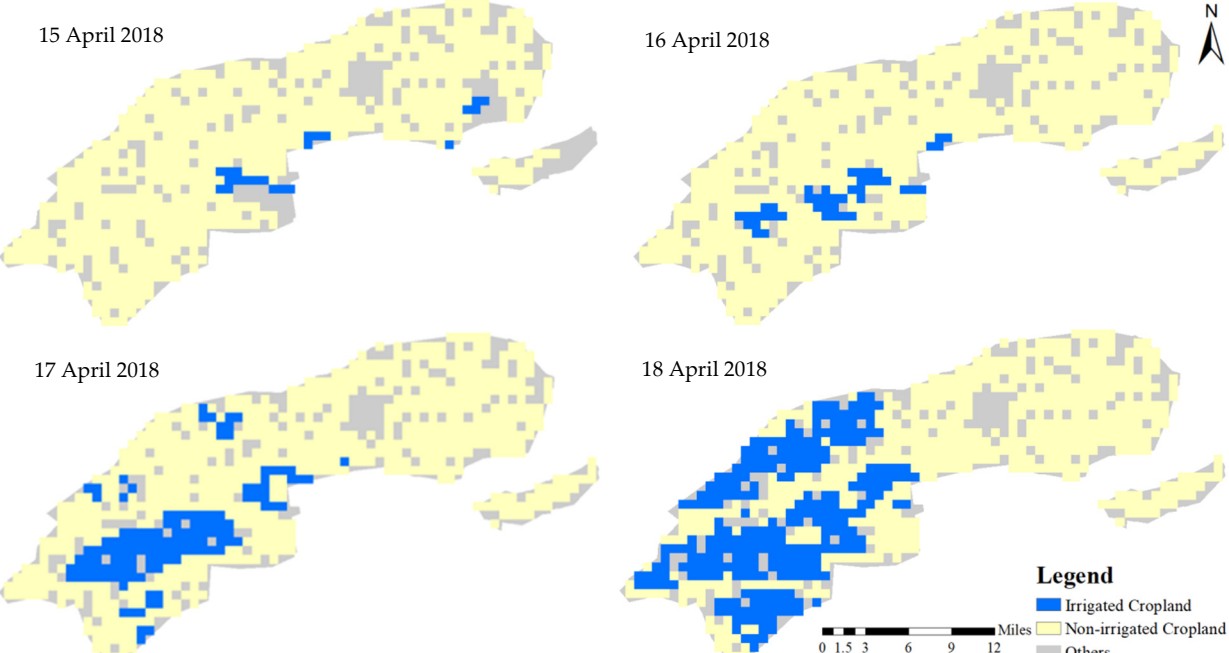

**Figure 9.** Spatial distribution of NIA from 15 April 2018 to 18 April 2018.

The spatial distribution of the irrigated area and irrigation intensity in the study area in the 8 years during the spring irrigation is shown in Figure 10. Because of the precipitation, cloud coverage, and other weather factors during the spring irrigation, the amount of available MODIS data was limited for 2012, 2014, and 2015. The MODIS data in 2011, 2013, and 2016–2018 were relatively good, and the available data could be used to identify the irrigation area. According to the precipitation conditions and crop demand, irrigation conditions varied between years (Table 2). Most of the croplands in the study area were irrigated once, some parts were irrigated twice, and a few parts were irrigated thrice. The smallest STIA was in 2013, accounting for 48.14% of the designed irrigated area. The largest STIA was in 2016, accounting for 88.93% of the designed irrigation area. The STIA accounted for 60–70% of the designed irrigation area in 2011, 2017, and 2018. Except for 2016, the SIA for the other four years was not very different and within the range of 55,900–73,100 ha. In 2016, irrigation began on 29 January and ended on 1 May, lasting 94 days, with the longest irrigation time from 2011 to 2018. Due to the drought in the region in 2016, the amount of crop irrigation increased, which significantly increased

the SIA to 100,200 ha in 2016. There has been more irrigation in the southwestern part of the study area since 2011. The areas with the most irrigated areas in the spring irrigation for 2011 and 2013 were the Jingyao, Liuqu, and Liuji Systems, while during the springs of 2016–2018, the Liuji, Pucheng, and Sunzhen Systems were the most irrigated. The largest SIA and STIA were observed in 2016 and covered almost all systems.

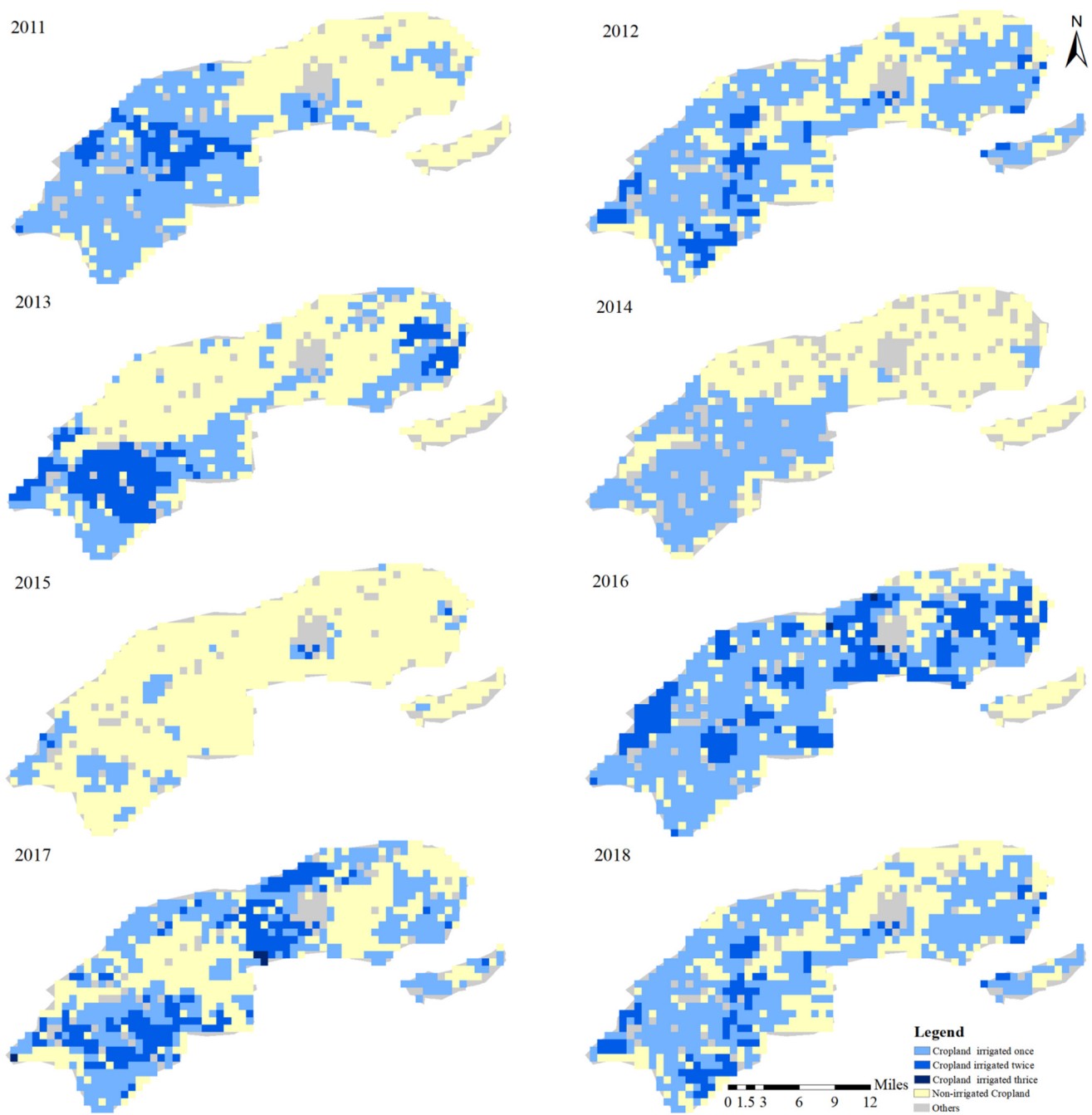

**Figure 10.** Irrigation intensity and spatial distribution of irrigation districts in spring irrigation in the study area from 2011 to 2018.

**Table 2.** Statistical table of SIA and STIA in the study area during spring irrigation from 2011 to 2018.

| Year | Data (Month/Day) | Area Irrigated Once (ha) | Area Irrigated Twice (ha) | Area Irrigated Thrice (ha) | SIA (ha) | STIA (ha) | Proportion of Total Designed Irrigated Area (%) |
|------|------------------|--------------------------|---------------------------|----------------------------|----------|-----------|------------------------------------------------|
| 2011 | 2/11–4/28 | 45,000 | 7900 | Null | 60,800 | 52,900 | 62.73 |
| 2012 | 3/2–3/13; 3/20–4/29 | 34,500 | 1500 | Null | 37,500 | 36,000 | 42.69 |
| 2013 | 2/14–3/20; 3/23–4/22 | 25,300 | 15,300 | Null | 55,900 | 40,600 | 48.14 |
| 2014 | 2/6–3/12; 3/23–4/22 | 34,400 | Null | Null | 34,400 | 34,400 | 40.79 |
| 2015 | 2/10–4/4 | 8900 | 600 | Null | 10,100 | 9500 | 11.26 |
| 2016 | 1/29–5/1 | 50,400 | 24,300 | 300 | 100,200 | 75,000 | 88.93 |
| 2017 | 1/20–4/22 | 38,900 | 16,500 | 400 | 73,100 | 55,800 | 66.17 |
| 2018 | 2/9–4/22 | 51,900 | 8500 | Null | 68,900 | 60,400 | 71.62 |

*4.4. Validation*

The proposed method is sensitive to changes in soil moisture content and can effectively identify regions with high soil moisture content. For example, light rain occurred in Fuping County for four days (3–6 March 2018) and Pucheng County only had a light rain on 4 March 2018. In the image of the study area for 8 March 2018, the irrigated area monitoring method based on DTR could be used to identify the regions with high soil water content caused by precipitation (Figure 11). The time interval between the last precipitation in Fuping County was short and the intensity was high, and most of the irrigated regions were detected. In Pucheng County, the interval between the last precipitation was long and the intensity was small, but the small areas of irrigation could still be detected. Therefore, the impact of precipitation needed to be removed before identifying the irrigation area.

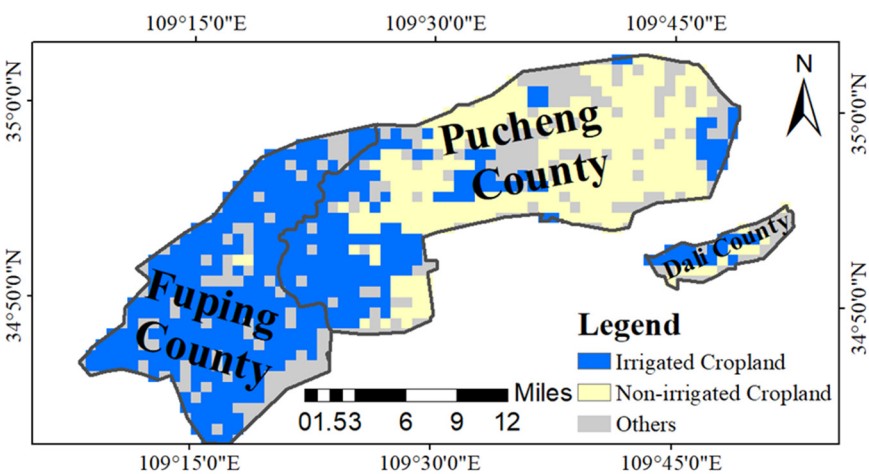

**Figure 11.** Spatial distribution map of the precipitation on 8 March 2018.

The monitoring results of the SIA and STIA in the study area were in line with the actual situation. The designed irrigated area of the study area was 84,333 ha, and the STIA by this method was smaller than the designed irrigated area, with a maximum of 75,000 ha (Table 2). In addition, comparing the SIA estimated by the statistical data (Table 3), the average monitoring accuracy of the SIA obtained by the above method was above 94% in 2011, 2013, 2016–2018. The available satellite remote sensing data volume was the main limiting factor for the accuracy of irrigated area monitoring results. Due to the small amount of available data, the irrigated area remote sensing monitoring effect was not satisfactory in 2012, 2014, and 2015. As shown in Figure 6, the DTR decreased after irrigation, but the DTR gradually increased with the evaporation of vegetation and topsoil, and it was not much different from that before irrigation on the third day after irrigation. If there are no data available within the first three days after irrigation, it is difficult to identify the irrigated area. So, the timeliness of the MOD11A1 data had a great impact on the accuracy

of the results. The amount of MOD11A1 data with good quality in the study area in 2012, 2014, and 2015 was less than 10 images, and other data were affected by cloud, rain and other factors resulting in null values. So, the DTR changes caused by irrigation were not captured, and the irrigation area could not be identified, resulting in a low accuracy in these three years. Similarly, due to the limitation of data quality, the monitoring results generally underestimated the irrigated area.

**Table 3.** Comparison of irrigated area monitoring results and statistical data in 2011–2018.

| Year | SIA (ha) | | Absolute Error (ha) | Accuracy (%) |
|------|-------------------|-----------------|---------------------|--------------|
| | **Monitoring Results** | **Statistical Data** | | |
| 2011 | 60,800 | 62,000 | −1200 | 98.06 |
| 2012 | 37,500 | 55,333 | −17,833 | 67.77 |
| 2013 | 55,900 | 64,667 | −8767 | 86.44 |
| 2014 | 34,400 | 67,240 | −32,840 | 51.16 |
| 2015 | 10,100 | 53,933 | −43,833 | 18.73 |
| 2016 | 100,200 | 106,847 | −6647 | 93.78 |
| 2017 | 73,100 | 74,733 | −1633 | 97.81 |
| 2018 | 68,900 | 66,333 | 2567 | 96.13 |

The irrigation area monitoring results of all systems in the study area were mostly consistent with the statistical data. Taking 2016 as an example, the monitoring results of SIA and the statistical data of all systems in the spring irrigation were compared, as shown in Table 4. Remote sensing monitoring results of irrigation systems such as Liuqu, Xingzhen, Sunzhen, Jingyao, Liuji, and Pucheng were 99.43%, 97.34%, 93.09%, 87.72%, 83.15%, and 72.15%, respectively, with a maximum error of less than 2600 ha. The crops in Dali System were mainly jujube trees in greenhouses, which made it hard to detect the changes in DTR before and after irrigation through satellite images, so the accuracy of irrigation area monitoring by proposed method in Dali System was relatively poor.

**Table 4.** Comparison of the SIA monitoring results and statistical data of each irrigation system in 2016.

| 2016 | Sunzhen System | Pucheng System | Jingyao System | Xingzhen System | Liuqu System | Liuji System | Dali System | Other |
|------|----------------|----------------|----------------|-----------------|--------------|--------------|-------------|-------|
| Extract SIA (ha) | 22,900 | 11,600 | 10,100 | 17,500 | 24,300 | 7400 | 200 | 6200 |
| Statistical data (ha) | 24,600 | 9073 | 11,513 | 17,047 | 24,440 | 8900 | 3727 | Null |
| Absolute error (ha) | −1700 | 2527 | −1413 | 453 | −140 | −1500 | −3527 | Null |
| Accuracy (%) | 93.09 | 72.15 | 87.72 | 97.34 | 99.43 | 83.15 | 5.37 | Null |

In addition, the spatial distribution of the remote sensing results of the irrigated districts based on DTR was also consistent with the actual situation. The study area mainly draws Yellow River water and uses the elevation difference to irrigate through the gravity flow channels. The Jingyao, Liuqu, and Liuji Systems have a low elevation and good irrigation conditions. The irrigated districts in Figure 10 are mainly distributed in patches, which is consistent with the channel irrigation characteristics. The main irrigation occurs once or twice during the spring, which is consistent with the ground-truth data.

We compared our results with the Global Map of Irrigation Area (GMIA) from FAO with a spatial resolution of 5 arc minutes or 0.083333 decimal degrees. The latest version of the GMIA is version 5, describing areas actually irrigated around the year 2005, which can be downloaded from FAO's website (https://www.fao.org/aquastat/en/geospatial-information/global-maps-irrigated-areas/latest-version/, accessed on 15 August 2022). The results showed that the STIA from GMIA in 2005 was 72,532 ha, which was larger than our results except 2016. Due to the limitation of data resolution, the residential areas in the study area were basically identified as irrigated by GMIA, overestimating the STIA.

The proposed method has a greater advantage to separate the irrigated area from the residential area.

## 5. Discussion

A remote sensing irrigated area monitoring method using LST and ancillary data is more suitable for dynamic monitoring where channel irrigation is used in semi-arid districts. Furthermore, the method can be used to obtain NIA, irrigation intensity, SIA, and STIA. In addition, the method can assist in the timely understanding of the distribution and changes of an irrigation area, which can be used to quantitatively understand the direction of irrigation water, drought resistance, and disaster relief, as well as providing technical support for the optimal allocation management of irrigation water resources.

There were significant differences and variations in the characteristics of the irrigation time schedule and spatial distribution with different irrigation sources and methods. Dynamic monitoring of irrigation processes using remote sensing requires a high timeliness and preferably a daily temporal resolution. The spatial resolution of remote sensing data should be based on comprehensive considerations such as the area of the monitoring district, planting structure, farming methods, and irrigation methods. Therefore, the application of remote sensing monitoring and the identification of irrigated area methods based on the DTR is affected by the temporal and spatial resolution of the remote sensing data. At present, the available remote sensing data sources of LST include MODIS, FY, and Landsat. The MOD11A1 LST product was selected in this study after considering data stability and availability. MOD11A1 data is time-sensitive and can obtain the daily LST data, both during the day and at night. However, cloud cover and other climate factors can mean that there are insufficient data to meet the requirements for dynamic irrigation monitoring in some cases.

China has a diverse climate, a complex agricultural planting structure, fragmented fields, and a low level of information regarding irrigation districts. Consequently, the cropland in one MODIS pixel (about 1 km$^2$) may be cultivated by dozens or even hundreds of households, and each household may irrigate their land at different times. If a small part of a pixel is irrigated, it may not be possible to detect irrigation in that pixel, and consequently the total irrigated area could be underestimated. Similarly, if most of the croplands in a pixel are irrigated, irrigation is detected for the whole pixel, and the total irrigated area could be overestimated. Therefore, the method can monitor the development of trends in irrigation, but it is not possible to accurately monitor the area of field irrigation because of the coarse spatial resolution of the imagery. Moreover, northwest China is arid and less rainy with larger DTR, which makes it suitable to use DTR for irrigated areas monitoring. Therefore, temperature changes caused by irrigation are easier to observe than that in the southern China. The applicability of this method in the southern China needs further study.

Combined with the irrigation regimes and the impact of precipitation, the monitoring results were processed and corrected. The NIA, irrigation intensity, SIA, and STIA of the channel irrigation method in the study area can be monitored according to the actual situation. However, the applicability of this method with distinct irrigation techniques, such as well control in smaller control districts, needs further study due to the spatial resolution of the data.

## 6. Conclusions

Taking Donglei Irrigation District (Phase II) as the study area, we proposed a more direct approach with a shorter cycle time to map irrigated areas during the spring at a 1-km spatial resolution with MODIS LST products and ancillary data for the historical meteorological data and main crop phenological characteristics. The distribution of irrigation intensity, SIA and STIA from 2011 to2018 was monitored by the proposed method in the study area, and the accuracy was evaluated by statistical data. The main conclusions that can be drawn from this work are as follows:

(1) The effect of irrigation measures on LST and its changes were analyzed through field experiments, and the feasibility of irrigated area mapping using LST was verified, which was used as a theoretical basis for studying the irrigated area mapping method based on LST. The experimental results showed that irrigation will reduce the DTR, and the extraction of the irrigated area can be achieved by delimiting the threshold of DTR;

(2) The spring irrigation in the study area was divided into three periods by historical meteorological data and the phenological characteristics of the main crops.: 16 January–23 February, 24 February–24 March, and 25 March–31 May. Based on the DTR obtained by MODIS from rain-fed plots and the historical meteorological data during the spring irrigation from 2011 to 2018, the threshold for each period was 12 °C, 15 °C, and 11 °C, respectively, using a mathematical statistical analysis method;

(3) After removing the data affected by precipitation, the NIA, SIA, and STIA in the study area from 2011 to 2018 were identified by using the thresholds. Most of the cropland was irrigated 1–2 times in spring for 2011–2018. The SIA was largest in 2016, accounting for 88.93% of the designed irrigated area. Except 2016, the SIA in the study area fluctuated within the range of 55,900–73,100 ha, and the STIA accounted for 60–70% of the designed irrigated area;

(4) This study used statistical data to evaluate the accuracy of remote sensing monitoring results. The average accuracy of SIA was above 94% when the amount of available data was sufficient. Moreover, the distribution of irrigation area for each irrigation system was consistent with the actual situation.

In areas where precipitation is low, it tends to be more concentrated in arid and semi-arid regions, and crop production in arid and semi-arid regions is often dependent on additional irrigation. Consequently, the change in vegetation cover after spring irrigation in these regions is more apparent. The soil water content changes before and after irrigation and the DTR changes are more obvious and easier to monitor and identify by remote sensing. The method proposed in this study had good applicability in arid and semi-arid regions for the periods of spring irrigation in the study area and in regions where the soil moisture content varied greatly before and after irrigation. Furthermore, this method is suitable for irrigation with a large control area, such as channel irrigation, as it is influenced by the timeliness and spatial resolution of LST remote sensing data. However, this method is greatly affected by climatic conditions, soil moisture content (before and after irrigation), and vegetation cover. Further study is needed to determine whether the method is applicable to other climatic zones (especially in humid regions) or other crops (e.g., rice or fruit trees).

**Author Contributions:** Y.L., W.S., L.T., X.C., R.G. and L.C. conceived the study. The main part of the research was completed by Y.L., W.S. and L.T. performed the experiments. X.C. provided language help and writing assistance for this paper. R.G. and L.C. participated in the data analysis for the study area. All authors have read and agreed to the published version of the manuscript.

**Funding:** This work was supported by the Jiangsu Water Conservancy Science and Technology Project (20211081) and the Special Projects for Basic Scientific Research of China Institute of Water Resources and Hydropower Research (JZ 01882204).

**Institutional Review Board Statement:** Not applicable.

**Informed Consent Statement:** Not applicable.

**Data Availability Statement:** The data presented in this study are available on request from the first author.

**Acknowledgments:** We would like to thank the Weinan Donglei Phase II Yellow River Engineering Administration, Weinan Donglei Yellowing Project Management Center for providing the research field and field working conditions.

**Conflicts of Interest:** The authors declare no conflict of interest.

## Abbreviations

| | |
|---|---|
| DTR | Diurnal Temperature Range |
| SIA | Spring Irrigated Areas |
| STIA | Total Irrigated Area in Spring |
| NIA | Near-real-time Irrigation Area |
| DLST | Daytime land surface temperature |
| NLST | Nighttime land surface temperature |
| FY | Fengyun satellite |

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
