# Peer review of "A New Method to Map Spring Irrigated Areas Using MODIS LST Products and Ancillary Data in an Agricultural District of Northwest China"

_water, doi:10.3390/w14172628_

Round 1

Reviewer 1 Report

L117-119 The quality of MODIS 117 data was good in 2011, 2016, 2017, and 2018, but the amount of available data was small 118 in 2012, 2014, and 2015 due to cloud coverage during spring irrigation

1) In what sense, the data quality was not good for specific years? Any evidence or supporting document?

2) What was the criteria for the quality control? Cloud? Give some specific details about the data quality

3) You may want to restrict that MODIS does not have good quality over specific study domain?

4) Revise it to “The quality of 117 MODIS data…”

Line 149-151 The statistical data obtained from the irrigation district management department includes the start and end times for spring irrigation in 2011–2018, the estimated SIA of the 150 study area ((where water consumption was at least 1,050 m3/ha), and the estimated SIA 151 of each system.

1) Please rewrite this sentence. Hard to understand

2) Remove the bracket (( à (

3) m3 à m3

Line 159 2:00 noon and 2:00 at night

à Is it local time? And change it to am and pm

Line 199 km2 à km2

Line 243 d à day

Line 280 rewrite the sentences

Results

Figure 8: Improve the quality of figure

Line 294-295 “the winter wheat plants are smaller, the vegetation coverage rate is lower, and there is more bare soil”

: I think smaller winter wheat plant (I guess it means height of vegetation) does not mean the low vegetation coverage. Author needs to rewrite this sentence

4.2 DTR Threshold determination

: I understand that authors put effort to bring up the rationale in the decision of DTR threshold. But, It would require sensitivity analysis with regards to the influence of DTR threshold variation in decision tree classification.

4.4 Validation

1) I strongly suggest authors to further verify the results of irrigated area by comparing with the Global Map of Irrigation Area (GMIA; FAO) for the further reference. I understand that author used the local authority’s dataset (maybe referred as statistical data) to verify the result while it would be also beneficial if you can verify that it showed strong advantage with the existing gridded reference.

2) Author may need to analyze the details of the absolute error and accuracy

- Provide the reason why 2015 showed significantly low accuracy. It cannot be just analyzed as MODIS 2015 datasets does not have good quality data.

- In addition, excluding the 2018, estimated irrigated area from this study continually showed underestimation. What is the reason for this phenomenon? It seems like author needs more detailed classification in decision tree

Table 4.

1) What is the reason why Dali System showed significant absolute error compared to the other region? Author needs to write it more carefully

Line 404: Low terrain means low elevation? Or homogeneous terrain?

Line 424: FY à need abbreviation

Line 424: Not all sentinel series provides the LST;

Line 425, 426 MODIS MOD11A1 à MOD11A1

Reviewer 2 Report

The manuscript is well written and illustrated. Some of my minor comments are as follows:

1. What are the reasons for increased SIA in 2016? There are not clear in the manuscript.

2. Further, please try to mention the reasons for the study being limited to 2018?? 

Round 2

Reviewer 1 Report

Thank you for author's effort and I can see that overall improvement of the submitted manuscript. However, there are still some questions need to be solved with regards to the newly added items

1. Sensitivity analysis  

   - Author mentioned that there are not significant change with respect to change of DTR threshold while that only accounts after the threshold after 10. It still shows significant change of STIA (ha) from 9 to 10 degree (700 ha to 32000 ha). Author still want to provide more rationale on selection of the thresholds.

  - Additionally, there are no information about the truth of STIA during the first period of spring irrigation in 2018. Author(s) only stated the SIA for 2/9-4/22 in the Table 2. It would be more informative if author can address the sensitivity of accuracy depending on the threshold. 

2. Part of discussion (Line 467-475).

  - Author made a discussion with regards to the diverse climate over China while this study only focused on northwest China. It would be great if author can associate the climate characteristics over northwest China with the experiment result, that would be much better.

3. English edit

- I still see some of typo as well as grammar error on the revised manuscript. Please carefully check English throughout the manuscript. 
